# Strategic SA-UNet: Integrating self-attention blocks into U-Net for efficient crack segmentation

Ryota Kobayashi[1], Munehiro Kimura[1], Ryosuke Harakawa[1], Norrima Mokhtar[2]*, Yang Zhou[2], Muhammad Amirul Aiman Asri[2], Raza Ali[3], Masahiro Iwahashi[1]

**1** Department of Electrical, Electronics and Information Engineering, Nagaoka University of Technology, Nagaoka, Niigata, Japan, **2** Human-Machine Interface Lab, Centre for Research in Industry 4.0, Faculty of Engineering, Universiti Malaya Lembah Pantai, Kuala Lumpur, Malaysia, **3** Department of Electrical Engineering, Faculty of Information and Communication Technology (FICT), Balochistan University of Information Technology, Engineering and Management Sciences (BUITEMS), Quetta, Pakistan

ォThese authors contributed equally to this work.
* norrimamokhtar@um.edu.my

## Abstract

Accurate crack segmentation plays a crucial role in ensuring safety and mitigating disaster risks during road inspections and structural health monitoring. However, traditional image processing techniques often struggle with low detection accuracy and poor generalization performance due to the diverse morphology of cracks and the presence of background noise. To address these challenges, MixSegNet, a model that combines the strengths of convolutional neural networks (CNNs) and Transformers, has been proposed and demonstrated to achieve high segmentation performance. However, this enhanced precision comes at the cost of prolonged training cycles, which limits its applicability in operational environments such as infrastructure inspection, where new data must be acquired and processed continuously and rapidly. In this paper, to address this limitation, we propose Strategic SA-UNet (Strategically Integrated Self-Attention U-Net), a novel crack segmentation network. The model strategically integrates a computationally efficient U-Net based CNN with a Self-Attention Block between the encoder and decoder to effectively fuse local features with global context, thereby maintaining high segmentation accuracy while reducing training time and computational cost. Experimental evaluations on publicly available datasets demonstrate that Strategic SA-UNet achieves segmentation accuracy comparable to MixSegNet, while reducing training time by 83%, Floating Point Operations (FLOPs) by 63%, and Model Parameters by 96%. Furthermore, Strategic SA-UNet achieves a high mean Intersection over Union (mIoU) even with a small number of epochs, highlighting its superior training efficiency. These results suggest that Strategic SA-UNet is an efficient segmentation model, especially suitable for real-time infrastructure inspection and structural monitoring applications.

**Data availability statement:** All relevant data used in this study are available from the Cracks-APCGAN dataset repository at https://github.com/tjboise/APCGAN-AttuNet.

**Funding:** This work was partially supported by Japan Society for the Promotion of Science (JSPS) KAKENHI Grant Number JP24K02975. The funders were involved in funding acquisition, supervision, and contributions to the writing (original draft, review, and editing).

**Competing interests:** The authors have declared that no competing interests exist.

## Introduction

Crack formation on pavement surfaces poses a significant safety risk for road users. The primary causes of pavement cracking include traffic load, construction quality, and environmental factors such as moisture and temperature [1,2]. The structural integrity of roads gradually deteriorates over time, and sustained traffic flow, particularly in urban areas, further exacerbates this degradation. A 2006 study reported that road condition-related accidents caused an estimated economic loss of $217.5 billion in the United States alone [3]. As road usage increases, so does the risk, sometimes leading to fatal accidents. Therefore, pavement maintenance is a top priority to ensure user safety. Furthermore, Ruchiyat et al. [4] demonstrated that the health of pavement infrastructure directly affects traffic efficiency and regional economic performance, while Ogbuehi et al. [5] reported that performing preventive maintenance when the pavement is still in good condition is significantly more cost-effective than repairing it after deterioration has progressed. These findings indicate that it is crucial to detect early signs of deterioration and take preventive actions before visible damage develops.

In pavement crack detection systems, line-scan or area-scan cameras are typically used to collect road surface images. For example, the digital highway data vehicle (DHDV) system is deployed on United States (US) highways for this purpose [6]. One promising approach to improve pavement maintenance is to use machine learning algorithms to segment cracks from the background. If sufficient accuracy is achieved, such models could serve as reliable alternatives to traditional visual inspection by pavement engineers. Traditional crack segmentation methods have relied heavily on image processing techniques such as thresholding [7] and mathematical morphology [8] in non-destructive testing (NDT). However, these methods are highly sensitive to image noise, including light reflections and shadows, and are therefore susceptible to false detections. Additionally, their limited adaptability to environmental changes such as lighting and weather conditions reduces their reliability in real-world applications.

To address this limitation, Yang et al. [9] proposed a novel approach based on a Fully Convolutional Network (FCN) [10], enabling the utilization of skeletonized cracks for detailed analyses of topological features, length, and width, which are critical indicators in practical evaluations. However, the shortage of training data for crack segmentation remains a significant challenge. To overcome this, Koenig et al. [11] developed an efficient learning strategy for the semantic segmentation of surface cracks. They adopted a patch-based training approach using a U-Net architecture [12], an FCN-based model, which achieved unprecedented results across various datasets with limited training samples. These studies demonstrate the potential of FCN-based methods to mitigate the data scarcity issue in crack segmentation.

Building on these advances, various U-Net derivatives, including PSNet [13], MSP U-Net [14], Rs-net [15], Mini-Unet [16], and AttuNet [17], have been proposed to further enhance local feature extraction and improve the accuracy of fine-structure detection. However, because all these architectures remain convolution-centric, they

struggle to capture the global context of an entire image, and pushing the networks deeper incurs sharply rising computational costs in both training and inference, creating an unfavourable trade-off.

To address this limitation, hybrid U-Net models incorporating Self-Attention Block mechanisms have emerged, including Swin-Unet [18], TransUNet [19], SETR [20], CrackFormer [21], CrackTrNet [22].

Furthermore, Zhou et al. [23] combined CNN and Transformer in a parallel architecture, enabling simultaneous utilization of CNN's strong local feature extraction capability and Transformer's global contextual understanding. This approach achieves high segmentation accuracy even in images with complex backgrounds. However, such a hybrid structure increases model complexity, and the incorporation of multiple additional modules, such as the Fuse Block and UC Block, leads to higher computational cost, leaving challenges for practical deployment.

Guijie et al. [24] tackled this by proposing RHA-Net, which combines Residual Blocks with Hybrid Attention Blocks and leverages depth-wise separable convolutions to achieve substantial model thinning. RHA-Net succeeds in slashing parameter counts and FLOPs while maintaining high accuracy, yet it still falls short of matching the performance of larger models, indicating that an optimal balance between efficiency and accuracy has not been reached.

In addition, recent research has proposed CCT Net [25], a model that integrates CNN and Transformer with practical deployment in mind and demonstrates both lightweight design and high crack segmentation accuracy compared to conventional methods. However, this model has not yet been sufficiently evaluated against large-scale models, so it is difficult to conclude that the balance between performance and efficiency has been fully verified.

In this paper, we introduce Strategic SA-UNet, which achieves comparable detection performance to MixSegNet [23] while drastically reducing training cost. Our model is built upon U-Net, known for its ability to learn effectively even from relatively small datasets. We focus on the common drawback in deep layers where spatial resolution becomes coarse, causing fine cracks to disappear. To address this, we strategically introduce a Self-Attention Block into the deeper layers, allowing the network to capture global contextual information, such as structural relationships and crack continuity across the entire image. This design compensates for the limitations of local features and enables the preservation of fine crack patterns while effectively reducing false detections.

The main contributions of this study are summarized as follows:

1. Strategic integration of a lightweight Self-Attention Block for accurate and efficient crack segmentation.

We propose a lightweight self-attention block inspired by the non-local block [26] and Self-Attention GAN [27], which is strategically inserted into the deeper encoder and early decoder layers of U-Net to enhance long-range feature dependency modelling. This enables the model to capture global contextual relationships in addition to local features, preserving thin crack continuity and reducing false detection. Experiments on the cracks-APCGAN [17] dataset demonstrated strong segmentation performance.

2. Faster convergence and reduced training cost with fewer epochs.

By appropriately placing Self-Attention Blocks in mid- and deep-layer stages, the balance between local and global representations is improved, resulting in faster convergence and fewer required training epochs compared to conventional methods. This is particularly beneficial for scenarios with limited computational resources.

The rest of this paper is structured as follows. In Materials and Methods, we describe the architecture of the proposed model and the details of the Self-Attention Block, as well as the experimental settings, including the computing environment, dataset, loss function, optimization method, and evaluation metrics. In Results and Discussion, we present both quantitative and qualitative segmentation results on benchmark datasets and compare the performance with existing methods. We also discussed the ablation study evaluating how the insertion position of the Self-Attention Block affects accuracy and computational cost, followed by limitations of this study and potential future research directions. Finally, in Conclusion, we summarize the contributions and findings of this work.

## Materials and methods

### Strategic SA-UNet

The overall architecture of the proposed Strategic SA-UNet for crack segmentation is illustrated in Fig 1. The model adopts a standard encoder–decoder structure with skip connections, enhanced by the integration of Self-Attention Blocks at intermediate and deeper layers. This design enables the network to capture both fine-grained local details and global contextual information, while maintaining computational efficiency.

In the encoder, each stage applies two 3×3 convolutional layers (padding = 1) with Rectified Linear Unit (ReLU) activations, followed by a 2×2 max-pooling layer with a stride of 2. This operation halves spatial resolution while doubling the number of feature channels, enabling hierarchical feature extraction and the construction of rich, high-dimensional representations from the input image.

The decoder restores spatial resolution using a 2×2 transposed convolution (up-convolution), which also halves the number of channels. The up-sampled feature map is then concatenated via skip connections with the correspondingly cropped encoder feature map, after which two additional 3×3 convolutions with ReLU activations are applied.

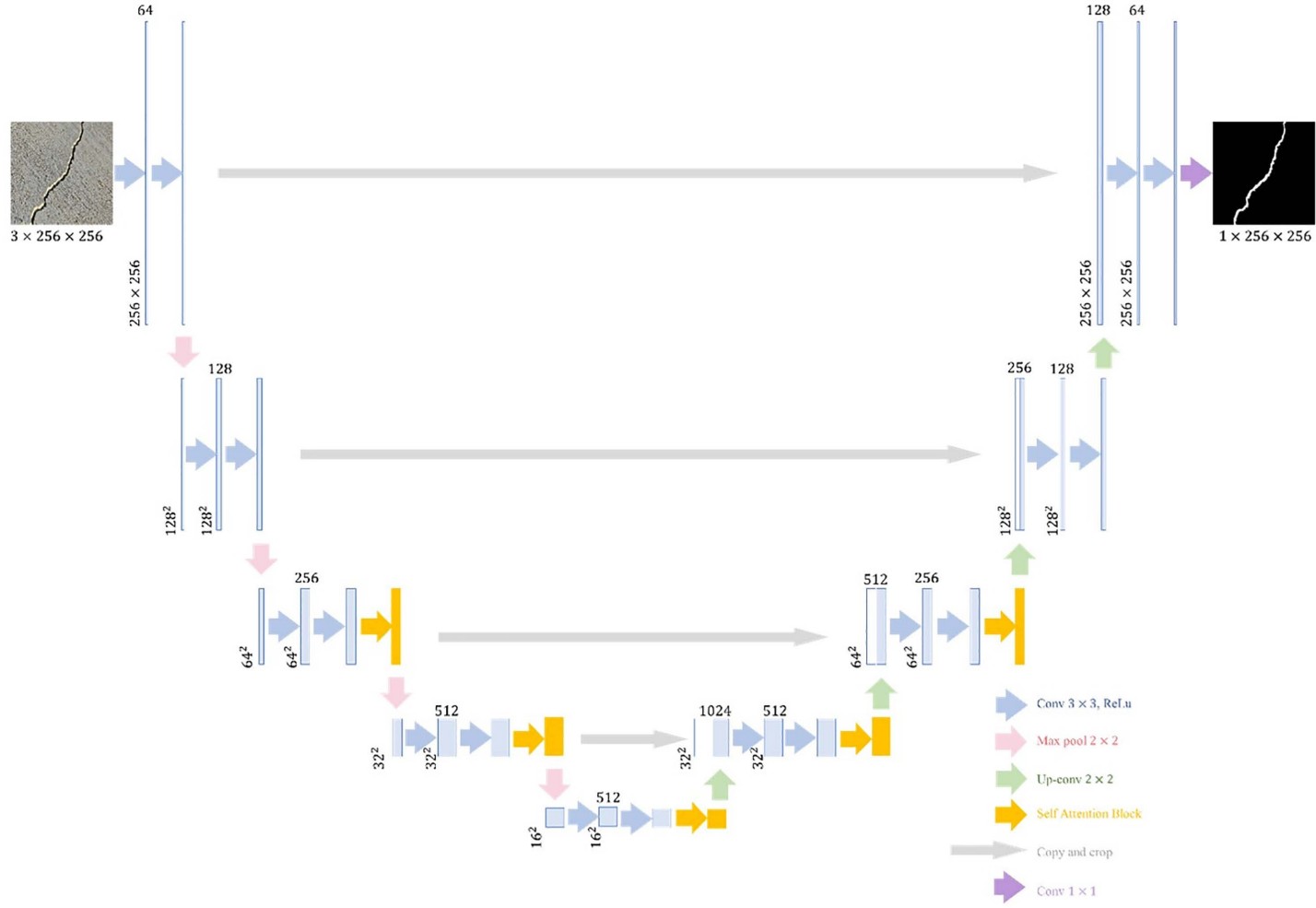

**Fig 1. The proposed Strategic SA-UNet framework.**

The cropping step compensates for the border pixels lost during convolution. Finally, a 1 × 1 convolution projects each 64-dimensional feature vector to the desired number of output classes.

We strategically insert Self-Attention Blocks into intermediate and deeper layers of both the encoder and decoder. This allows the network to learn long-range dependencies between distant regions of the image, thereby integrating global contextual information alongside local features. As a result, Strategic SA-UNet achieves improved detection of elongated and fragmented crack patterns and suppresses background noise and false positives, all while keeping additional computational cost to a minimum.

### Self-attention block

It consists of three main components that generate the Query ($Q$), Key ($K$), and Value ($V$) matrices from the input feature map. These are derived through separate 1 × 1 convolutional layers. To improve learning efficiency, the number of channels in $Q$ and $K$ is reduced to one-eighth of the channel dimension, while V retains the original number of channels to preserve feature richness. The spatial relationships between different positions in the input feature map are learned through the inner product of $Q$ and $K$. The resulting attention map is normalized using the Softmax function, as shown in Eq (1):

$$Attention\ Map = Softmax\left(Q \cdot K^T\right) \tag{1}$$

This attention map is then applied to $V$, enabling a weighted aggregation of spatial features, emphasizing the most relevant regions. A learnable scaling parameter $\gamma$ is introduced to control the contribution of the attention output. Finally, a residual connection is added to preserve the original input features while integrating the attention-enhanced representation, as described in Eq (2):

$$Output = \gamma \cdot Attention\ Map \cdot V + Input \tag{2}$$

The Self-Attention Block effectively reduces computational overhead by downscaling the dimensions of $Q$ and $K$, and is selectively applied to specific layers in the network to maximize performance gains while minimizing training cost.

### Implementation

In this study, the proposed method was implemented using Python on Google Colab Pro +, utilizing GPU acceleration. All image processing tasks were executed in a virtual environment equipped with an NVIDIA A100 GPU (40 GB VRAM) and 83.5 GB of system RAM.

### Dataset

In this study, we selected cracks-APCGAN [17], a secondary open-source dataset, as the benchmark dataset. This dataset is based on the publicly available DeepCrack dataset [28] and has been extended with additional images, making it a valuable resource for crack segmentation research. The cracks-APCGAN dataset was created by enhancing the original training set using APCGAN, a GAN-based data augmentation technique. In this method, new images similar to the original dataset are generated using a generative adversarial network (GAN), and then manually annotated to ensure label accuracy. The enhanced dataset, as shown in the original study, has been demonstrated to significantly improve the training process, making it an ideal choice for our work. The dataset used in this study comprised 585 crack images (256 × 256 pixels) for training and 237 crack images (256 × 256 pixels) for evaluation (testing). In addition to the cracks-APCGAN dataset used for training, we employed the CrackForest dataset as an independent test set to evaluate generalization performance. The CrackForest dataset is publicly available on Kaggle and consists of 118 crack images captured under different environmental conditions, including variations in pavement texture, lighting, and background appearance. Unlike

cracks-APCGAN, which is augmented using GAN-based techniques, CrackForest contains real-world crack images without synthetic augmentation. Therefore, using CrackForest solely for testing allows us to examine whether models trained on augmented data can generalize unseen real-world distributions.

## Loss function

In this study, we adopt the Dice loss [29] to improve the accuracy of crack segmentation. While cross-entropy loss is commonly used in segmentation tasks, it is highly sensitive to class imbalance, where the background class typically dominates over the crack class. To address this issue, Focal loss [30] has been proposed, which focuses on hard-to-classify samples. However, Focal Loss does not explicitly consider spatial structure or continuity, making it less effective in capturing fine and thin crack patterns.

In contrast, Dice loss maximizes the overlap between the predicted and ground truth masks, inherently mitigating class imbalance. This characteristic makes it particularly suitable for segmenting narrow and elongated structures like cracks. By focusing on the accuracy of the foreground region, Dice loss enables precise and reliable segmentation performance for crack detection tasks.

## Optimizer

In this study, the RMSprop optimization algorithm was employed to shorten training time and achieve faster convergence for the crack segmentation task. The widely used AdamW optimizer [31], a variant of Adam with an improved weight decay strategy, is known for enhancing generalization performance. However, as shown in Fig 2, when combined with Dice loss, AdamW required more than 600 epochs to achieve sufficient convergence. Therefore, we selected RMSprop to accelerate training and improve convergence speed. Based on prior experience and empirical observations, the initial learning rate was set to $1 \times 10^{-5}$. This value was chosen to balance convergence speed and training stability, preventing excessively large update steps in the early stages of training, which could otherwise destabilize the model. To ensure a fair comparison, the number of training epochs was fixed at 200 for all models, allowing us to evaluate both segmentation performance and computational cost under consistent conditions.

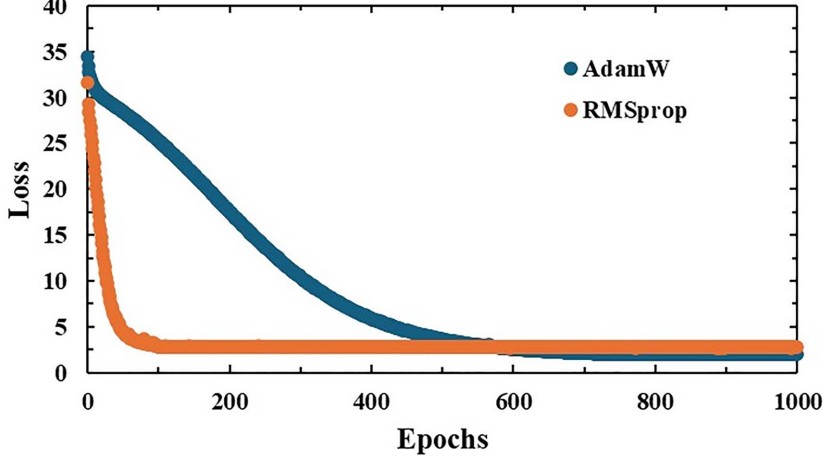

**Fig 2. Convergence of the loss function during training of the Strategic SA-UNet model.**

## Evaluation metrics

To establish the basis for model evaluation, the definitions of true positives, false positives, true negatives, and false negatives used in crack segmentation are summarized in Table 1. These definitions serve as the foundation for the evaluation metrics employed in this study. In this study, we employ these metrics to quantitatively evaluate the performance of the proposed crack segmentation model.

In crack segmentation, Precision refers to the proportion of correctly identified crack pixels among all pixels predicted as cracks by the model. This metric evaluates the accuracy of the model's positive predictions and is calculated using Eq (3): Here, the summation symbol ($\sum$) denotes the total number of pixels across the entire dataset.

$$Precision = \frac{\sum p}{\sum p + \sum \overline{p}} \tag{3}$$

Recall indicates the proportion of actual crack pixels that are correctly identified by the model. This metric evaluates the sensitivity of the model concerning the ground truth positive regions and is calculated using Eq (4):

$$Recall = \frac{\sum p}{\sum p + \sum \overline{g}} \tag{4}$$

The F1 score is the harmonic mean of Precision and Recall, serving as a balanced metric that combines both measures. It provides a comprehensive evaluation of the model's accuracy and sensitivity. A higher F1 score indicates better overall performance. The F1 score is calculated using Eq (5):

$$F1 = \frac{2 * Precision * Recall}{Precision + Recall} \tag{5}$$

In semantic segmentation, mean Intersection over Union (mIoU) is a metric that represents the average IoU (Intersection over Union) across all classes. It is calculated as the ratio of the overlap between the predicted region and the ground truth region to the union of both areas. A higher mIoU value indicates better segmentation accuracy. The mIoU is computed using Eq (6):

$$mIoU = \frac{1}{C} \sum_{C=1}^{C} \frac{\sum p}{\sum p + \sum \overline{p} + \sum \overline{g}} \tag{6}$$

## Computational complexity and memory profiling

To quantify the computational cost of our models, we measured Floating Point Operations (FLOPs) using the FlopCountAnalysis utility from the fvcore library. With the network in evaluation mode, we passed a single dummy tensor of size

**Table 1. Definitions in crack segmentation.**

|  | Prediction Crack | Prediction Non-Crack |
|---|---|---|
| Ground Truth Crack | $p$ (*True Positive*) | $\overline{g}$ (*False Negative*) |
| Ground Truth Non-Crack | $\overline{p}$ (*False Positive*) | $g$ (*True Negative*) |

(1 × 3 × 256 × 256) through the model and recorded the total number of multiply accumulate operations. The resulting count was divided by $10^9$ to report GFLOPs.

Peak GPU memory consumption was profiled during training by invoking PyTorch's torch.cuda.max_memory_allocated() API, which returns the maximum number of bytes allocated on the device, has been converted to megabytes for reporting purposes. Both GFLOPs and peak Video Random Access Memory (VRAM) usage are reported alongside segmentation accuracy to enable a fair comparison of computational efficiency across different architectures.

The inference speed (FPS) was evaluated by measuring the average processing time over 100 forward passes with a batch size of 1, using a single dummy tensor of size (1 × 3 × 256 × 256). FPS was computed as the inverse of the average inference time per image.

## Results and discussions

### Computational efficiency and model complexity analysis

To benchmark the computational efficiency of Strategic SA-UNet, its training and inference characteristics were compared with those of U-Net, AttuNet, and MixSegNet under identical training conditions. All models were trained on the cracks-APCGAN dataset using the same hardware and optimization settings. Table 2 summarizes the comparison in terms of training time, FLOPs, model parameters, VRAM consumption, and FPS.

Table 2 summarizes the training and inference efficiency of Strategic SA-UNet in comparison with U-Net, AttuNet, and MixSegNet under identical training conditions. Strategic SA-UNet requires a training time of 0.6003 h, which is slightly longer than that of U-Net (0.5341 h) and AttuNet (0.5867 h), but dramatically shorter than that of MixSegNet (3.625 h).

In terms of computational complexity, Strategic SA-UNet requires only 40.12 G FLOPs and 14.24 M parameters, representing substantial reductions compared with MixSegNet (109.2 G FLOPs and 309.7 M parameters). The slightly longer training time compared to U-Net and AttuNet can be attributed to the incorporation of Self-Attention Block modules, which introduce multiple matrix multiplications and Softmax operations. Although these attention-based operations exhibit relatively low FLOPs, Softmax normalization in particular involves substantial data transfer between GPU global memory and compute units, resulting in low arithmetic intensity. As reported in recent transformer optimization studies [32], such operations are often memory-bandwidth bound, which explains the increased training time despite the reduced FLOPs.

From the perspective of inference efficiency, Strategic SA-UNet requires 7.99 GB of VRAM, which is higher than that of U-Net (4.56 GB) and AttuNet (4.65 GB), and achieves an inference speed of 142.2 FPS, which is lower than that of U-Net (240.3 FPS) and AttuNet (156.2 FPS). However, compared with MixSegNet, which consumes 13.94 GB of VRAM and operates at only 22.6 FPS, Strategic SA-UNet maintains high segmentation accuracy while achieving a practical balance between memory consumption and inference speed.

**Table 2. Training and inference efficiency comparison of crack segmentation models under identical training conditions. (Best values in bold. Lower values indicate better performance for training time, FLOPs, model parameters, and VRAM usage, while higher values indicate better performance for FPS.).**

| Dataset | cracks-APCGAN | | | | |
|---|---|---|---|---|---|
| Metric | Training time [h] | FLOPs | Model Parameters | VRAM [GB] | FPS [frames/s] |
| Unet | **0.5341** | 55.81 G | 31.39 M | **4.556** | **240.3** |
| Attunet | 0.5867 | 66.54 G | 34.88 M | 4.650 | 156.2 |
| MixSegNet | 3.625 | 109.2 G | 309.7 M | 13.940 | 22.64 |
| SA-Unet | 0.6003 | **40.12 G** | **14.24 M** | 7.988 | 142.2 |

Overall, these results demonstrate that Strategic SA-UNet offers a superior trade-off among segmentation accuracy, computational efficiency, and model compactness, making it well suited for deployment in environments with limited computational resources.

## Comparison of segmentation performance of each model

To benchmark the effectiveness of Strategic SA-UNet, its performance was compared with U-Net, AttuNet, and MixSegNet under identical training conditions. All models were trained on the cracks-APCGAN dataset and evaluated on both the cracks-APCGAN test set and the CrackForest dataset. Table 3 presents a comparative evaluation using Precision, Recall, F1 score, and mIoU as accuracy metrics.

As shown in Table 3, Strategic SA-UNet consistently achieves the best overall performance across both datasets. On the cracks-APCGAN test set, Strategic SA-UNet outperforms MixSegNet by +0.27% in Precision, + 0.33% in Recall, and +0.50% in mIoU. Compared with U-Net and AttuNet, Strategic SA-UNet also shows consistent improvements in Precision (+0.70%/ + 0.59%), Recall (+1.03%/ + 0.82%), and mIoU (+1.45%/ + 1.19%). These results indicate that Strategic SA-UNet enhances segmentation accuracy while maintaining stable performance on in-distribution data.

More importantly, Strategic SA-UNet demonstrates superior generalization performance on the CrackForest dataset, which was not used during training. It achieves the highest mIoU of 60.28%, along with Precision of 89.54%, Recall of 61.53%, and an F1-score of 72.94%. Compared with MixSegNet, Strategic SA-UNet improves Precision by +0.54%, Recall by +0.69%, F1-score by +0.66%, and mIoU by +0.62%. These gains can be attributed to the ability of Strategic SA-UNet to preserve local feature extraction in shallow layers while strategically inserting Self-Attention Blocks in intermediate and deep layers to effectively capture global contextual information.

## Qualitative comparison of crack segmentation results

To further validate the quantitative results, a qualitative comparison of crack segmentation performance was conducted across multiple test images. Fig 3(a)–3(f) illustrates seven representative segmentation results randomly selected from a variety of scenes. The figure includes the original input images, corresponding ground truth masks, and the outputs generated by Strategic SA-UNet, MixSegNet, U-Net, and AttuNet, allowing a direct visual comparison of their ability to detect fine cracks, branching points, and background noise.

As observed in Fig 3, Strategic SA-UNet was able to trace thin, branching cracks and their bifurcation points without interruption, and produced almost no false positives in the background. These findings confirm that Strategic SA-UNet simultaneously achieves fine-structure preservation and false-positive suppression at a high level, reproducing crack morphology with the greatest accuracy. Moreover, by incorporating Self-Attention Blocks to leverage global contextual information effectively, it maintains high-quality segmentation.

## Training convergence analysis

To analyze the training behavior and convergence characteristics of each model, the mIoU metric was continuously monitored over the full range of training epochs. Fig 4 illustrates the evolution of mIoU values for Strategic SA-UNet,

**Table 3. Generalization performance of crack segmentation models on cracks-APCGAN and CrackForest datasets. (Best values in bold.).**

| Dataset | cracks-APCGAN | | | | CrackForest | | | |
|---------|---------------|--|--|--|-------------|--|--|--|
| Metric | Precision [%] | Recall [%] | F1 score [%] | mIoU [%] | Precision [%] | Recall [%] | F1 score [%] | mIoU [%] |
| Unet | 97.22 | 90.7 | 93.85 | 88.73 | 89.4 | 58.06 | 70.4 | 57.06 |
| Attunet | 97.33 | 90.91 | 94.01 | 88.99 | 89.51 | 58.39 | 70.68 | 57.38 |
| MixSegNet | 97.65 | 91.4 | 94.42 | 89.68 | 89 | 60.84 | 72.27 | 59.66 |
| SA-Unet | **97.92** | **91.73** | **94.72** | **90.18** | **89.54** | **61.53** | **72.94** | **60.28** |

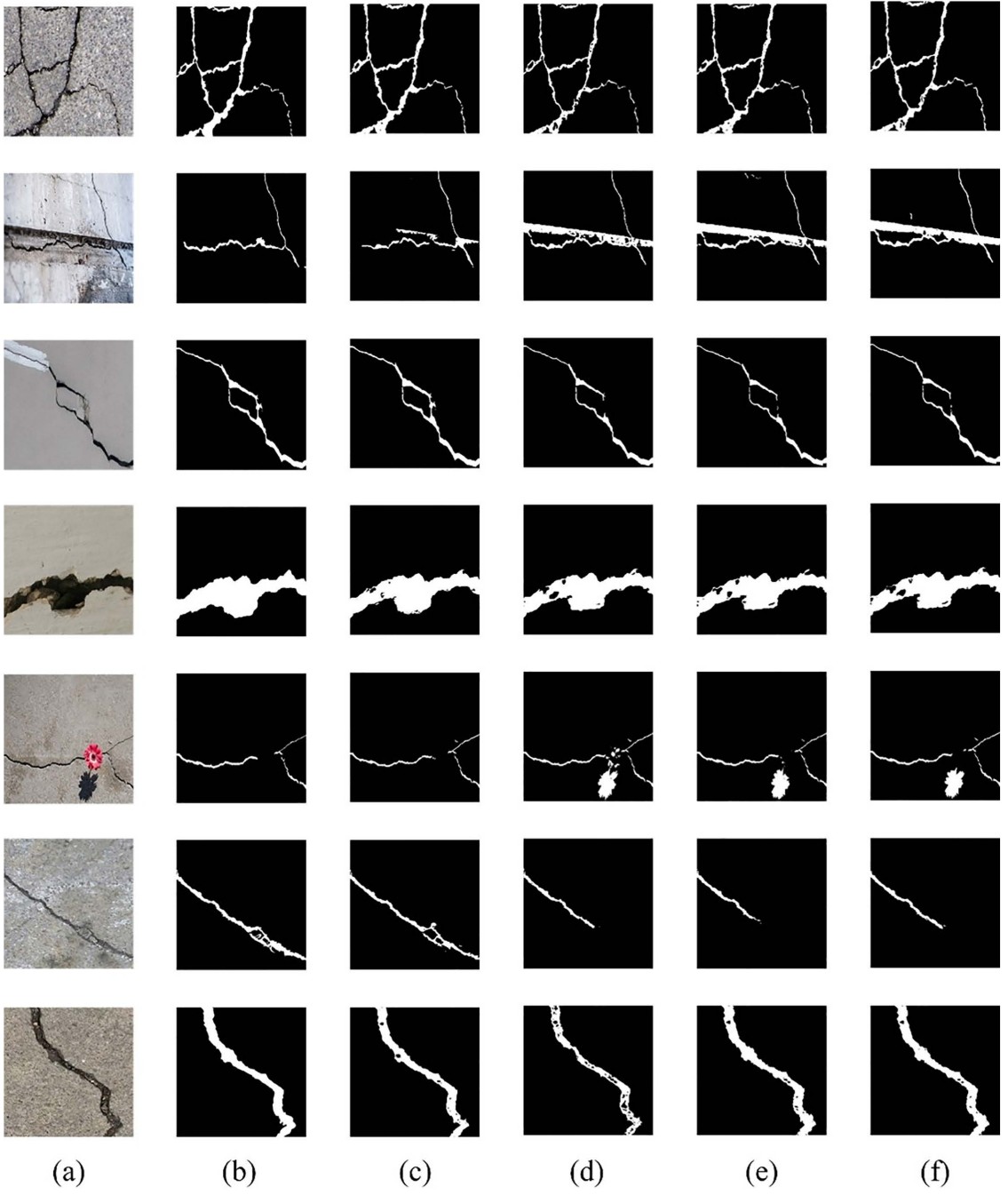

**Fig 3. A comparison of the results obtained using different segmentation models in various scenarios.** (a) Original Image (b) Ground Truth (c) Strategic SA-UNet (d) MixSegNet (e) U-Net (f) AttuNet.

MixSegNet, U-Net, and AttuNet, providing a direct comparison of their learning progress, stability, and consistency. This visualization highlights how each architecture responds to training, including any fluctuations or plateaus in performance, and allows a clearer assessment of convergence efficiency.

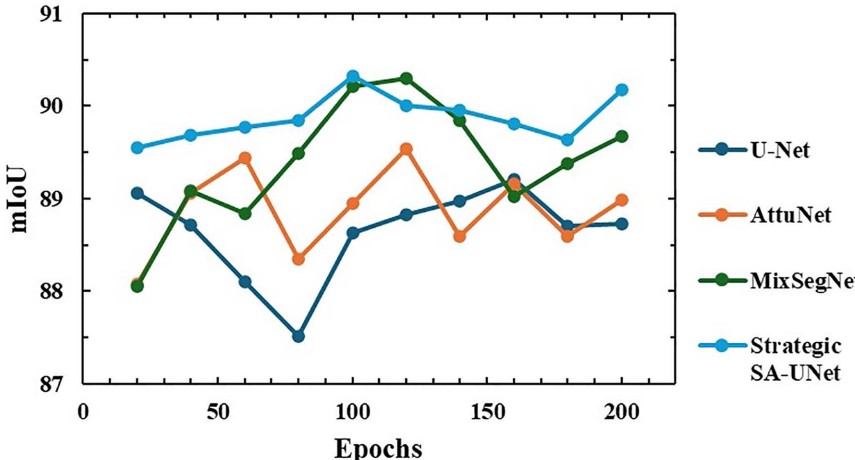

**Fig 4. mIoU variation with training epochs for each model.**

As observed in Fig 3, Strategic SA-UNet was able to trace thin, branching cracks and their bifurcation points without interruption, and produced almost no false positives in the background. These findings demonstrate that Strategic SA-UNet concurrently achieves high-level preservation of fine structural details and suppression of false positives, thereby reproducing crack morphology with superior accuracy. Moreover, by incorporating Self-Attention Blocks to leverage global contextual information effectively, it maintains high-quality segmentation.

Strategic SA-UNet consistently outperforms the others throughout training, even in the early stages (epochs 50–100). This underscores the benefit of selectively integrating Self-Attention blocks into the intermediate and deep layers of both the encoder and decoder, allowing SA-Unet to capture local features and global context from the outset. By contrast, Transformer-based architectures like MixSegNet deliver promising results but generally require more data and longer training to converge fully, which can be a limitation when resources are constrained. U-Net and AttuNet exhibit larger fluctuations in mIoU and lower average accuracy compared to Strategic SA-UNet and MixSegNet. Although AttuNet applies attention in its skip connections, its limited view of the global context enhances local details while constraining its capability to model complex, global structural patterns.

## Ablation study

To further investigate how Self-Attention Block placement influences model behaviour, an ablation study was conducted on Strategic SA-UNet and its variants. Table 4 summarizes the impact of inserting the Self-Attention Block at different network

**Table 4. Comparison of model performance on test data.**

| Models | Precision [%] | Recall [%] | F1 [%] | mIoU [%] | Training time [h] | FLOPs | Model Parameters |
|---|---|---|---|---|---|---|---|
| Base One | 97.43 | 90.21 | 93.68 | 88.41 | 0.5707 | **31.01 G** | **13.40 M** |
| First layer | 97.79 | 91.31 | 94.44 | 89.69 | 0.5850 | 38.68 G | 13.50 M |
| Second layer | 97.27 | 90.82 | 93.93 | 88.87 | 0.5398 | 32.33 G | 13.80 M |
| Third layer | 97.11 | 91.30 | 94.11 | 89.21 | **0.5363** | 31.13 G | 13.72 M |
| Strategic SA-UNet | **97.92** | **91.73** | **94.72** | **90.18** | 0.6003 | 40.12 G | 14.24 M |

depths (first, second, and third layers) on segmentation performance and computational cost. This analysis provides insight into how feature-level context aggregation at various layers affects both accuracy metrics and resource requirements.

The baseline model ("Base One") without any Self-Attention Block achieved a Precision of 97.43%, a Recall of 90.21%, an F1 score of 93.68%, and an mIoU of 88.41%. Its training time was 0.5707 h, with 31.01 GFLOPs and 13.40 M parameters. While this baseline provides a solid starting point, both F1 and mIoU metrics were lower than in attention-augmented variants, indicating that Self-Attention Block can indeed enhance performance.

When a Self-Attention Block was inserted in the first (shallowest) layer, the model reached a Precision of 97.79%, a Recall of 91.31%, an F1 score of 94.44%, and an mIoU of 89.69%. Training time increased slightly to 0.5850 h, and computational cost rose to 38.68 GFLOPs with 13.50 M parameters. This shallow-layer configuration produced the most significant individual gains over the baseline, improving F1 by 0.76% and mIoU by 1.28%, as it emphasizes low-level features early in the network. However, it still lagged the full SA-UNet, likely due to limited global context awareness.

Placing the attention block in the second (intermediate) layer yielded a Precision of 97.27%, a Recall of 90.82%, an F1 score of 93.93%, and an mIoU of 88.87%. Interestingly, Recall increased by 0.51% compared to the first-layer variant, while Precision declined by 0.52%. Training required only 0.5398 h, with 32.33 GFLOPs and 13.80 M parameters. This suggests that intermediate-level attention broadens feature capture but may introduce sensitivity to local noise.

When the block was positioned in the third (deepest) layer, the model recorded a Precision of 97.11%, a Recall of 91.30%, an F1 score of 94.11%, and an mIoU of 89.21%. Its training time was 0.5363 h, with 31.13 GFLOPs and 13.72 M parameters. While Recall remained comparable to the first-layer placement, Precision decreased further, indicating that deep-layer attention enhances global context integration at the expense of fine-grained detail.

Finally, the full Strategic SA-UNet model distributes Self-Attention Blocks across multiple encoder and decoder layers, achieving a Precision of 97.92%, a Recall of 91.73%, an F1 score of 94.72%, and an mIoU of 90.18%. This peak performance required 0.6003 h of training, with 40.12 GFLOPs and 14.24 M parameters. Although training time increased by 0.0296 h and 2.11 G compared to the baseline, the substantial improvements in F1 and mIoU justify these additional costs.

Although the results are promising, several limitations still need to be addressed. The proposed Strategic SA-UNet successfully achieves high segmentation accuracy while significantly reducing computational cost. Although the model was trained primarily on the publicly available cracks-APCGAN dataset and its generalization capability was partially validated using an unseen dataset, CrackForest, the diversity of real-world conditions remains limited. In particular, scenarios such as nighttime scenes, rain, shadows, surface stains, and variations in pavement materials have not yet been sufficiently covered. Therefore, future work will involve evaluating the model on more diverse and complex real-world pavement datasets to further assess and improve its robustness and generalization capability.

Second, Strategic SA-UNet demonstrates a favorable balance between segmentation accuracy and model compactness compared to large-scale models such as MixSegNet. While the incorporation of Self-Attention Blocks inevitably introduces additional computational overhead during training compared to purely CNN-based architectures such as U-Net and AttuNet, this increase represents a design trade-off for enhanced global feature modeling rather than a limitation of the proposed approach. To further improve training efficiency without sacrificing performance, future work will explore meta-heuristic optimization strategies, such as virus-spread-inspired algorithms [33], to adaptively tune hyperparameters and stabilize the training process. Furthermore, inspired by recent lightweight architectures such as MHAED-Net [34], we aim to refine the structure of the Self-Attention Block and optimize the overall network design, enabling Strategic SA-UNet to maintain accuracy comparable to large-scale models like MixSegNet while further reducing parameters and FLOPs, thereby facilitating efficient deployment on edge devices.

## Conclusion

In this study, we proposed Strategic SA-UNet, a lightweight crack segmentation model in which Self-Attention Blocks are strategically placed in the mid and deep layers of both the encoder and decoder. By selectively integrating attention, the

model effectively balances local texture feature extraction with global contextual understanding, achieving high segmentation accuracy with low computational cost. The main findings of this study are summarized as follows:

1. Significant reduction in training cost while maintaining high accuracy. Strategic SA-UNet achieved a Precision of 97.92%, Recall of 91.73%, F1 score of 94.72%, and mIoU of 90.18% on the cracks-APCGAN dataset. Compared with MixSegNet, the proposed model reduced training time by approximately 83% and FLOPs by 63%, while maintaining a lightweight architecture with only 14.24 M parameters and outperforming other methods.

2. Generalization Performance and Deployment Efficiency Experiments conducted on a completely unseen dataset confirmed that Strategic SA-UNet maintains accuracy comparable to MixSegNet, while achieving superior inference speed and lower memory consumption. These results suggest that the proposed method exhibits strong generalization capability across different data distributions and can be practically deployed in environments with limited computational resources.

3. Superior convergence and training stability. From the mIoU transition analysis, Strategic SA-UNet exhibited smooth convergence and maintained high accuracy from the early training stage (50–100 epochs). The model demonstrated smaller performance fluctuations than U-Net and AttU-Net, and faster convergence compared to MixSegNet.

4. Effectiveness of the strategic placement of Self-Attention Blocks. Ablation studies confirmed that distributing Self-Attention Blocks across multiple layers yields the best performance, improving F1 by +1.04% and mIoU by +1.77% compared with non-attention configurations.

## Author contributions

**Conceptualization:** Ryota Kobayashi, Norrima Mokhtar, Raza Ali.

**Data curation:** Yang Zhou, Raza Ali.

**Formal analysis:** Ryota Kobayashi, Ryosuke Harakawa, Norrima Mokhtar.

**Funding acquisition:** Masahiro Iwahashi.

**Investigation:** Ryota Kobayashi, Muhammad Amirul Aiman Asri.

**Methodology:** Ryota Kobayashi, Muhammad Amirul Aiman Asri.

**Project administration:** Norrima Mokhtar.

**Resources:** Norrima Mokhtar, Yang Zhou.

**Software:** Ryota Kobayashi.

**Supervision:** Munehiro Kimura, Ryosuke Harakawa, Norrima Mokhtar, Raza Ali, Masahiro Iwahashi.

**Validation:** Ryota Kobayashi, Norrima Mokhtar.

**Visualization:** Ryota Kobayashi.

**Writing – original draft:** Ryota Kobayashi.

**Writing – review & editing:** Munehiro Kimura, Ryosuke Harakawa, Norrima Mokhtar, Muhammad Amirul Aiman Asri, Masahiro Iwahashi.

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
