## [Decision Letter · Decision Letter 0]

23 Nov 2025

Dear Dr. Mokhtar,

Thank you for submitting your manuscript to PLOS ONE. After careful consideration, we feel that it has merit but does not fully meet PLOS ONE’s publication criteria as it currently stands. Therefore, we invite you to submit a revised version of the manuscript that addresses the points raised during the review process.

We look forward to receiving your revised manuscript.

Kind regards,

Yuk Ming Tang

Academic Editor

PLOS ONE

Journal Requirements:

“Japan Society for the Promotion of Science (JSPS) KAKENHI Grant Number JP24K02975”

Please state what role the funders took in the study. If the funders had no role, please state: 'The funders had no role in study design, data collection and analysis, decision to publish, or preparation of the manuscript.'

Additional Editor Comments:

Please clarifying the strategic integration of self-attention blocks and broaden experiments to diverse datasets. Additionally, address limitations and future work directions

Reviewers' comments:

Reviewer's Responses to Questions

**Comments to the Author**

1. Is the manuscript technically sound, and do the data support the conclusions?

Reviewer #1: Yes

Reviewer #2: Yes

2. Has the statistical analysis been performed appropriately and rigorously?

Reviewer #1: Yes

Reviewer #2: Yes

3. Have the authors made all data underlying the findings in their manuscript fully available?

Reviewer #1: Yes

Reviewer #2: Yes

4. Is the manuscript presented in an intelligible fashion and written in standard English?

Reviewer #1: Yes

Reviewer #2: Yes

Reviewer #1: 1- The research contributions should be clearly and concisely outlined in bullet points, expressing the techniques or tools used to achieve each one.

2- Adding a paragraph at the end of the introduction section that explains the structure of the research in all its sections

3- Providing a review of related works, highlighting key research in the field published over the past five years, to discuss the advantages and disadvantages of each technology.

4- Comparing the submitted work with research published in 2024 and 2025

5- Comparing the work presented by comparing the results using the ablation system

6- Adding at least three important references in the world by 2025

Reviewer #2: This paper proposes Strategic SA-UNet, a lightweight U-Net variant integrating self-attention blocks to improve crack segmentation efficiency. While the model achieves notable reductions in training time (83%) and FLOPs (63%) compared to MixSegNet, its contributions are incremental. Some important results have been obtained. Still, the minor modifications or clarifications should be made before the possible acceptance.

(1) The language of this paper is good, and can be further improved.

(2) If possible, please clarify why self-attention is integrated strategically (e.g., hierarchical feature fusion) rather than uniformly.

(3) If possible, please broader experiments: Test on diverse datasets (e.g., CrackTree, CFD) and compare with more recent architectures.

(4) If possible, please report inference time, memory usage, and parameter efficiency on edge devices.

(5) If possible, please address limitations (e.g., dataset bias) and future work (e.g., multi-modal fusion for occluded cracks).

(6) The necessary recent reports (https://doi.org/10.1007/s11227-025-07456-8,
https://doi.org/10.1088/1361-6501/adbf3c,
https://doi.org/10.1016/j.measurement.2024.116210) on the lightweight multiscale hybrid attention encoder-decoder network, and image segmentation technology for industrial application can be cited to support the experimental findings and discussions.

(7) SECTION Conclusion needs to be refined and reduced. The important experimental findings and theoretical conclusions can be list by 3-5 points.

**Do you want your identity to be public for this peer review?** For information about this choice, including consent withdrawal, please see our Privacy Policy

Reviewer #1: No

Reviewer #2: No

---

## [Author Response · Author response to Decision Letter 1]

6 Jan 2026

To Editor,

1) Please clarifying the strategic integration of self-attention blocks and broaden experiments to diverse datasets. Additionally, address limitations and future work directions.

MixSegNet achieves high accuracy by combining a CNN and a Transformer architecture. However, its large model size makes it less suitable for lightweight inference required in practical deployment scenarios. In response to this issue, we designed our model based on U-Net, which can be effectively trained even with a relatively small amount of data. Nevertheless, U-Net tends to lose fine crack details in deeper layers due to reduced spatial resolution. To overcome this limitation, we strategically integrate Self-Attention Blocks into the mid and deep encoder layers. This design enables the model to acquire global contextual information that complements local features by capturing long-range dependencies and structural continuity across the entire image. As a result, the proposed method preserves thin crack connectivity and suppresses false detection tasks that are challenging for conventional CNN based architectures while achieving accuracy comparable to MixSegNet with improved training efficiency and reduced computational cost. (Page 5, Line 14)

Next, we expanded the scope of experimental validation to evaluate the generalization capability of the proposed method. In the previous manuscript, the effectiveness of the proposed approach on the cracks-APCGAN dataset was demonstrated in Page 15 and Table 3. Based on these results, and considering the practical constraint that collecting and annotating sufficient real-world data is often difficult in practical applications, we conducted additional tests using a completely unseen dataset that was not used during training to examine the generality and applicability of the proposed method. The test dataset used in this evaluation was CrackForest, which is publicly available on Kaggle. A description of the CrackForest dataset was added on Page 9, Line 16. The corresponding results were newly added to Table 2 on Page 14 and Table 3 on Page 15 in the revised manuscript. The results show that Strategic SA-UNet maintains performance comparable to MixSegNet, while achieving superior efficiency in terms of both VRAM consumption and inference time. Furthermore, the above detailed analyses of training and inference efficiency, including training time, FLOPs, model parameters, VRAM consumption, and FPS, were added to the revised manuscript on Page 13, Line 14 and Page 14, Line 4. In addition, performance comparisons across both the cracks-APCGAN and CrackForest datasets were newly added to the revised manuscript on Page 15, Lines 8 and 17, thereby further clarifying the computational efficiency, effectiveness, and generalization capability of Strategic SA-UNet.

Finally, we clarified the limitations of this study and outlined future research directions. First, although the proposed method was trained mainly on the publicly available cracks-APCGAN dataset and its generalization capability was partially validated using the unseen CrackForest dataset, the diversity of real-world conditions remains limited. In particular, scenarios such as nighttime scenes, rain, shadows, surface contamination, and variations in pavement materials have not yet been sufficiently covered. Therefore, future work will involve experiments on more diverse and complex real-world pavement datasets to further evaluate and improve the robustness and generalization of the model. Second, inspired by recent lightweight architectures, we aim to refine the structure of the Self-Attention Block and optimize the overall network design. This will enable Strategic SA-UNet to maintain accuracy comparable to large-scale models such as MixSegNet while further reducing the number of parameters and FLOPs, thereby facilitating efficient deployment on edge devices. These revisions have been added to Page 20, Line 12 in the revised manuscript.

To Reviewer #1,

1 - The research contributions should be clearly and concisely outlined in bullet points, expressing the techniques or tools used to achieve each one.

The main contributions of this study are summarized as follows:

1. Strategic integration of a lightweight Self-Attention Block for accurate and efficient crack segmentation.

We propose a lightweight self-attention block inspired by the non-local block [26] and Self-Attention GAN [27], which is strategically inserted into the deeper encoder and early decoder layers of U-Net to enhance long-range feature dependency modelling. This enables the model to capture global contextual relationships in addition to local features, preserving thin crack continuity and reducing false detection. Experiments on the cracks-APCGAN [17] dataset demonstrated strong segmentation performance.

2. Faster convergence and reduced training cost with fewer epochs.

By appropriately placing Self-Attention Blocks in mid- and deep-layer stages, the balance between local and global representations is improved, resulting in faster convergence and fewer required training epochs compared to conventional methods. This is particularly beneficial for scenarios with limited computational resources.

2 - Adding a paragraph at the end of the introduction section that explains the structure of the research in all its sections.

The rest of this paper is structured as follows. In Materials and Methods, we describe the architecture of the proposed model and the details of the Self-Attention Block, as well as the experimental settings, including the computing environment, dataset, loss function, optimization method, and evaluation metrics. In Results and Discussion, we present both quantitative and qualitative segmentation results on benchmark datasets and compare the performance with existing methods. We also discussed the ablation study evaluating how the insertion position of the Self-Attention Block affects accuracy and computational cost, followed by limitations of this study and potential future research directions. Finally, in Conclusion, we summarize the contributions and findings of this work.

3 - Providing a review of related works, highlighting key research in the field published over the past five years, to discuss the advantages and disadvantages of each technology.

Based on the reviewer’s suggestion, in addition to the description of RHA-Net at Page 4, Line 17, we added the following descriptions at Page 4, Line 21 and Page 5, Line 8.

Furthermore, Zhou et al. [23] combined CNN and Transformer in a parallel architecture, enabling simultaneous utilization of CNN’s strong local feature extraction capability and Transformer’s global contextual understanding. This approach achieves high segmentation accuracy even in images with complex backgrounds. However, such a hybrid structure increases model complexity, and the incorporation of multiple additional modules, such as the Fuse Block and UC Block, leads to higher computational cost, leaving challenges for practical deployment.

In addition, recent research has proposed CCT Net [25], a model that integrates CNN and Transformer with practical deployment in mind and demonstrates both lightweight design and high crack segmentation accuracy compared to conventional methods. However, this model has not yet been sufficiently evaluated against large-scale models, so it is difficult to conclude that the balance between performance and efficiency has been fully verified.

4 - Comparing the submitted work with research published in 2024 and 2025.

Similar to the above comment, this study discusses both the advantages and limitations of MixSegNet published in 2024 and CCT Net reported in 2025, thereby clarifying the positioning of the proposed model with respect to closely related recent works.

5 - Comparing the work presented by comparing the results using the ablation system.

Ablation studies have been already conducted in the previous manuscript (Page 18, Lines 12) to evaluate the contribution of the Self-Attention Block under different placement strategies. Specifically, we compared a baseline model without attention and several variants with the block inserted at different levels. The results demonstrate that while shallow- and deep-layer attention improves performance individually, the best result is obtained when Self-Attention Blocks are strategically distributed across multiple encoder and decoder layers. The full Strategic SA-UNet achieved the highest scores (Precision: 97.92%, Recall: 91.73%, F1: 94.72%, mIoU: 90.18%), confirming the effectiveness of our design.

6 - Adding at least three important references in the world by 2025.

We added three important references published in 2025 that discuss the causes of pavement cracking, its impacts on infrastructure performance, and the associated economic and maintenance costs at Page 3, Lines 3 and 9, thereby strengthening the explanations of the causes and impacts of pavement cracking as described below.

Crack formation on pavement surfaces poses a significant safety risk for road users. The primary causes of pavement cracking include traffic load, construction quality, and environmental factors such as moisture and temperature [1,2]

Furthermore, Ruchiyat et al. [4] demonstrated that the health of pavement infrastructure directly affects traffic efficiency and regional economic performance, while Ogbuehi et al. [5] reported that performing preventive maintenance when the pavement is still in good condition is significantly more cost-effective than repairing it after deterioration has progressed. These findings indicate that it is crucial to detect early signs of deterioration and take preventive actions before visible damage develops.

(Added references to the revised manuscript)

2. Jiang J, Xu K, Song Y, Gao L, Zhang J. Investigation of cracking behavior in asphalt pavement using digital image processing technology. Front. Built Environ. 2025;11:1580379. doi:10.3389/fbuil.2025.1580379.

4. Ruchiyat F, Prabowo YS. Cost-benefit analysis of rigid pavement road construction using exponential, NPV, and ERR methods (case study: Cimanying–Jiput road section, Pandeglang, Banten). Jurnal Teknik Sipil Cendekia. 2025;6(2):374–392. doi:10.51988/jtsc.v6i2.342.

5. Ogbuehi DD, Nnaji GO, Achukee CK, Mkparu DE. Optimization of road pavement maintenance in Nigeria. World Scientific News. 2025;200:190–204.

To Reviewer #2,

(1) The language of this paper is good, and can be further improved.

≫A native speaker reviewed the language of the revised manuscript and corrected wording as necessary before submission.

Page 7, Line 8:

【Original term】 ReLU

【Revised term】 Rectified Linear Unit (ReLU)

Page 7, Line 10:

【Original statement】 This operation halves the spatial resolution while doubling the number of feature channels,

【Revised statement】 This operation halves spatial resolution while doubling the number of feature channels,

(2) If possible, please clarify why self-attention is integrated strategically (e.g., hierarchical feature fusion) rather than uniformly.

We added the following explanation to Page 5, Line 14 in the revised manuscript:

MixSegNet achieves high accuracy by combining CNN and Transformer architectures; however, its large model size makes it less suitable for lightweight inference in practical deployment scenarios. To address this issue, we designed our model based on U-Net, which can be trained effectively even with relatively small datasets. However, the conventional U-Net often loses fine crack details in deeper layers due to reduced spatial resolution. To mitigate this issue, we strategically integrate Self-Attention Blocks into the mid- and deep encoder layers, allowing the model to capture global contextual information that complements local features by modeling long-range dependencies and structural continuity across the image. Consequently, the proposed method preserves thin crack connectivity, reduces false detections, and achieves comparable accuracy to MixSegNet while improving training efficiency and reducing computational cost.

(3) If possible, please broader experiments: Test on diverse datasets (e.g., CrackTree, CFD) and compare with more recent architectures.

We expanded the scope of experimental validation to evaluate the generalization capability of the proposed method. In the previous manuscript, the effectiveness of the proposed approach on the cracks-APCGAN dataset was demonstrated in Page 15 and Table 3. Based on these results, and considering the practical constraint that collecting and annotating sufficient real-world data is often difficult in practical applications, we conducted additional tests using a completely unseen dataset that was not used during training to examine the generality and applicability of the proposed method. The test dataset used in this evaluation was CrackForest, which is publicly available on Kaggle. A description of the CrackForest dataset was added on Page 9, Line 16. The corresponding results were newly added to Table 2 on Page 14 and Table 3 on Page 15 in the revised manuscript. The results show that Strategic SA-UNet maintains performance comparable to MixSegNet, while achieving superior efficiency in terms of both VRAM consumption and inference time. Furthermore, the above detailed analyses of training and inference efficiency, including training time, FLOPs, model parameters, VRAM consumption, and FPS, were added to the revised manuscript on Page 13, Line 14 and Page 14, Line 4. In addition, performance comparisons across both the cracks-APCGAN and CrackForest datasets were newly added to the revised manuscript on Page 15, Lines 8 and 17, thereby further clarifying the computational efficiency, effectiveness, and generalization capability of Strategic SA-UNet.

The latest architectures for road crack detection are primarily based on CNN-based architectures (i.e., MixSegNet), and a comparison with such approaches was already conducted in the previous manuscript. For example, newer architectures beyond CNNs, such as diffusion models, have recently been proposed, and their effectiveness has been reported in other research fields. Diffusion-based models may also have potential for road crack detection. However, implementing and systematically comparing such models would require additional technical development of new methodologies. Therefore, these investigations are beyond the scope of the present paper and are left as future work.

(4) If possible, please report inference time, memory usage, and parameter efficiency on edge devices.

≫All experiments were conducted using an NVIDIA A100 GPU with 40 GB of VRAM and 83.5 GB of system RAM via Google Colab Pro+. However, within the scope of this revision, direct access to edge devices was limited, therefore, measurements on actual edge hardware are our future work. Instead, we report the inference speed (FPS) and peak GPU memory usage (VRAM) measured under a unified experimental setup, which has been added to Table 2 on Page 14 in the revised manuscript. These metrics are widely used as effective proxy indicators for evaluating deployment feasibility in resource-constrained edge environments. In addition, a description of the FPS evaluation protocol was added to Page 13, Line 8 in the revised manuscript, as described below:

The inference speed (FPS) was evaluated by measuring the average processing time over 100 forward passes with a batch size of 1, using a single dummy tensor of size (1 × 3 × 256 × 256). FPS was computed as the inverse of the average inference time per image.

(5) If possible, please address limitations (e.g., dataset bias) and future work (e.g., multi-modal fusion for occluded cracks).

≫ We clarified the limitations of this study and outlined future research directions. First, although the proposed method was trained mainly on the publicly available cracks-APCGAN dataset and its generalization capability was partially validated using the unseen CrackForest dataset, the diversity of real-world conditions remains limited. In particular, scenarios such as nighttime scenes, rain, shadows, surface contamination, and variations in pavement materials have not yet been sufficiently covered. Therefore, future work will involve exp

---

## [Decision Letter · Decision Letter 1]

2 Feb 2026

Strategic SA-UNet: Integrating Self-Attention Blocks into U-Net for Efficient Crack Segmentation

PONE-D-25-54350R1

Dear Dr. Mokhtar,

We’re pleased to inform you that your manuscript has been judged scientifically suitable for publication and will be formally accepted for publication once it meets all outstanding technical requirements.

Kind regards,

Yuk Ming Tang

Academic Editor

PLOS One

Additional Editor Comments (optional):

Reviewers' comments:

Reviewer's Responses to Questions

**Comments to the Author**

Reviewer #1: (No Response)

Reviewer #2: All comments have been addressed

2. Is the manuscript technically sound, and do the data support the conclusions?

Reviewer #1: (No Response)

Reviewer #2: Yes

3. Has the statistical analysis been performed appropriately and rigorously?

Reviewer #1: (No Response)

Reviewer #2: Yes

4. Have the authors made all data underlying the findings in their manuscript fully available?

Reviewer #1: (No Response)

Reviewer #2: Yes

5. Is the manuscript presented in an intelligible fashion and written in standard English?

Reviewer #1: (No Response)

Reviewer #2: Yes

Reviewer #1: (No Response)

Reviewer #2: The authors have carefully addressed the review comments. Based on the overall quality of this manuscript, I think this paper can be accepted.

**Do you want your identity to be public for this peer review?** For information about this choice, including consent withdrawal, please see our Privacy Policy

Reviewer #1: No

Reviewer #2: No

---

## [Editor Report · Acceptance letter]

PONE-D-25-54350R1

PLOS One

Dear Dr. Mokhtar,

I'm pleased to inform you that your manuscript has been deemed suitable for publication in PLOS One. Congratulations! Your manuscript is now being handed over to our production team.

Kind regards,

on behalf of

Dr. Yuk Ming Tang

Academic Editor

PLOS One